# Marginal Policy Gradients: A Unified Family of Estimators for Bounded Action Spaces with Applications

**Carson Eisenach**[1,*,†,‡], **Haichuan Yang**[2,*,†], **Ji Liu**[2,3,†], and **Han Liu**[4,†]

[1]Department of Operations Research and Financial Engineering, Princeton University, Princeton, NJ 08544.
[2]Department of Computer Science, University of Rochester, Rochester, NY 14627.
[3]Kwai AI Lab at Seattle, Seattle, WA.
[4]Department of Electrical Engineering and Computer Science, Northwestern University, Evanston, IL 60208.

## Abstract

Many complex domains, such as robotics control and real-time strategy (RTS) games, require an agent to learn a continuous control. In the former, an agent learns a policy over $\mathbb{R}^d$ and in the latter, over a discrete set of actions each of which is parametrized by a *continuous* parameter. Such problems are naturally solved using policy based reinforcement learning (RL) methods, but unfortunately these often suffer from high variance leading to instability and slow convergence. Unnecessary variance is introduced whenever policies over bounded action spaces are modeled using distributions with unbounded support by applying a transformation $T$ to the sampled action before execution in the environment. Recently, the variance reduced clipped action policy gradient (CAPG) was introduced for actions in bounded intervals, but to date no variance reduced methods exist when the action is a direction, something often seen in RTS games. To this end we introduce the angular policy gradient (APG), a stochastic policy gradient method for *directional control*. With the *marginal policy gradients* family of estimators we present a unified analysis of the variance reduction properties of APG and CAPG; our results provide a stronger guarantee than existing analyses for CAPG. Experimental results on a popular RTS game and a navigation task show that the APG estimator offers a substantial improvement over the standard policy gradient.

## 1 Introduction

Recent work in deep reinforcement learning (RL) has achieved human level-control for complex tasks like Atari 2600 games and the ancient game of Go. Mnih et al. (2015) show that it is possible to learn to play Atari 2600 games using end to end reinforcement learning. Other authors (Silver et al., 2014) derive algorithms tailored to continuous action spaces, such as appear in problems of robotics control. Today, solving RTS games is a major open problem in RL (Foerster et al., 2016; Usunier et al., 2017; Vinyals et al., 2017); these are more challenging than previously solved game domains because the action and state spaces are far larger. In RTS games, actions are no longer chosen from a relatively small discrete action set as in other game types. Neither is the objective solely learning a continuous control. Instead the action space typically consists of many discrete actions each of which has a continuous parameter. For example, a discrete action in an RTS game might be moving the player controlled by the agent with a parameter specifying the movement direction. Because the agent must learn a continuous parameter for each discrete action, a policy gradient method is a natural approach to an RTS game. Unfortunately, obtaining stable, sample-efficient performance from policy gradients remains a key challenge in model-free RL.

---

[*]These authors contributed equally.
[†]This work was done while at the Tencent AI Lab, Bellevue, WA 98004.
[‡] Correspondence to: `eisenach@princeton.edu`.

Just as robotics control tasks often have actions restricted to a bounded interval, Multi-player Online Battle Arena (MOBA) games, an RTS sub-genre, often have actions restricted to the unit sphere which specify a direction (e.g. to move or attack). The current practice, despite most continuous control problems having bounded action spaces, is to use a Gaussian distribution to model the policy and then apply a transformation $T$ to the action $a$ before execution in the environment. This support mismatch between the *sampling action distribution* (i.e. the policy $\pi$), and the *effective action distribution* can both introduce bias to and increase the variance of policy gradient estimates (Chou et al., 2017; Fujita & Maeda, 2018). For an illustration of how the distribution over actions $a$ is transformed under $T(a) = a/||a||$, see Figure 1 in Section 3.

In this paper, motivated by an application to a MOBA game, we study policy gradient methods in the context of directional actions, something unexplored in the RL literature. Like CAPG for actions in an interval $[\alpha, \beta]$, our proposed algorithm, termed *angular policy gradient* (APG), uses a variance-reduced, unbiased estimated of the true policy gradient. Since the key step in APG is an update based on an estimate of the policy gradient, it can easily be combined with other state-of-the art methodology including value function approximation and generalized advantage estimation (Sutton et al., 2000; Schulman et al., 2016), as well as used in policy optimization algorithms like TRPO, A3C, and PPO (Schulman et al., 2015; Mnih et al., 2016; Schulman et al., 2017).

Beyond new methodology, we also introduce the *marginal policy gradients* (MPG) family of estimators; this general class of estimators contains both APG and CAPG, and we present a unified analysis of the variance reduction properties of all such methods. Because marginal policy gradient methods have already been shown to provide substantial benefits for clipped actions (Fujita & Maeda, 2018), our experimental work focuses only on angular actions; we use a marginal policy gradient method to learn a policy for the 1 vs. 1 map of the *King of Glory* game and the *Platform2D-v1* navigation task, demonstrating improvement over several baseline policy gradient approaches.

## 1.1 RELATED WORK

**Model-Free RL.** Policy based methods are appealing because unlike value based methods they can support learning policies over discrete, continuous and parametrized action spaces. It has long been recognized that policy gradient methods suffer from high variance, hence the introduction of trust region methods like TRPO and PPO (Schulman et al., 2015; 2017). Mnih et al. (2016) leverage the independence of asynchronous updating to improve stability in actor-critic methods. See Sutton & Barto (2018) for a general survey of reinforcement learning algorithms, including policy based and actor-critic methods. Recent works have applied policy gradient methods to parametrized action spaces in order to teach an agent to play RoboCup soccer (Hausknecht & Stone, 2016; Masson et al., 2016). Formally, a parametrized action space $\mathcal{A}$ over $K$ discrete, parametrized actions is defined as $\mathcal{A} := \bigcup_k \{(k, \omega) : \omega \in \Omega_k\}$, where $k \in [K]$ and $\Omega_k$ is the parameter space for the $k^{th}$ action. See Appendix B.5 for rigorous discussion of the construction of a distribution over parametrized action spaces and the corresponding policy gradient algorithms.

**Bounded Action Spaces.** Though the action space for many problems is bounded, it is nonetheless common to model a continuous action using the multivariate Gaussian, which has unbounded support (Hausknecht & Stone, 2016; Florensa et al., 2017; Finn et al., 2017). Until recently, the method for dealing with this type of action space was to sample according to a Gaussian policy and then either (1) allow the environment to clip the action and update according to the unclipped action or (2) clip the action and update according to the clipped action (Chou et al., 2017). The first approach suffers from unnecessarily high variance, and the second approach is off-policy.

Recent work considers variance reduction when actions are clipped to a bounded interval (Chou et al., 2017; Fujita & Maeda, 2018). Depending upon the way in which the $Q$-function is modeled, clipping has also been shown to introduce bias (Chou et al., 2017). Previous approaches are not applicable to the case when $T$ is the projection onto the unit sphere; in the case of clipped actions, unlike previous work, we do not require that each component of the action is independent and obtain much stronger variance reduction results. Concurrent work (Fellows et al., 2018) also considers angular actions, but their method cannot be used as a drop in replacement in state of the art methods and the a special form of the critic $q_\pi$ is required.

**Integrated Policy Gradients.** Several recent works have considered, as we do, exploiting an integrated form of policy gradient (Ciosek & Whiteson, 2018; Asadi et al., 2017; Fujita & Maeda,

2018; Tamar et al., 2012). Ciosek & Whiteson (2018) introduces a unified theory of policy gradients, which subsumes both deterministic (Silver et al., 2014) and stochastic policy gradients (Sutton et al., 2000). They characterize the distinction between different policy gradient methods as a choice of quadrature for the expectation. Their Expected Policy Gradient algorithm uses a new way of estimating the expectation for stochastic policies. They prove that the estimator has lower variance than stochastic policy gradients. Asadi et al. (2017) propose a similar method, but lack theoretical guarantees. Fujita & Maeda (2018) introduce the clipped action policy gradient (CAPG) which is a partially integrated form of policy gradient and provide a variance reduction guarantee, but their result is not tight. By viewing CAPG as a marginal policy gradient we obtain tighter results.

**Variance Decomposition.** The law of total variance, or variance decomposition, is given by $\text{Var}[Y] = \mathbb{E}[\text{Var}(Y|X)] + \text{Var}[\mathbb{E}[Y|X]]$, where $X$ and $Y$ are two random variables on the same probability space. Our main result can be viewed as a special form of law of total variance, but it is highly non-trivial to obtain the result directly from the law of total variance. Also related to our approach is Rao-Blackwellization (Blackwell, 1947) of a statistic to obtain a lower variance estimator.

## 2 PRELIMINARIES

**Notation and Setup.** For MDP's we use the standard notation. $\mathcal{S}$ is the state space, $\mathcal{A}$ is the action space, $p$ denotes the transition probability kernel, $p_0$ the initial state distribution, $r$ the reward function. A policy $\pi(a|s)$ is a distribution over actions given a state $s \in \mathcal{S}$. A sample trajectory under $\pi$ is denoted $\tau_\pi := (s_0, a_0, r_1, s_1, a_1, \dots)$ where $s_0 \sim p_0$ and $a_t \sim \pi(\cdot|s_t)$. The state-value function is defined as $v_\pi(s) := \mathbb{E}_\pi[\sum_{t=0}^\infty \gamma^t r_{t+1}|s_0 = s]$ and the action-value function as $q_\pi(s, a) := \mathbb{E}_\pi[\sum_{t=0}^\infty \gamma^t r_{t+1}|s_0 = s, a_0 = a]$. The objective is to maximize expected cumulative discounted reward, $\eta(\pi) = \mathbb{E}_{p_0}[v_\pi(s_0)]$. $\rho_\pi$ denotes the improper discounted state occupancy distribution, defined as $\rho_\pi := \sum_t \gamma^t \mathbb{E}_{p_0}[\mathbb{P}(s_t = s|s_0, \pi)]$. We make the standard assumption of bounded rewards.

We consider the problem of learning a policy $\pi$ parametrized by $\boldsymbol{\theta} \in \Theta$. All gradients are with respect to $\boldsymbol{\theta}$ unless otherwise stated. By convention, we define $0 \cdot \infty = 0$ and $\frac{0}{0} = 0$. A measurable space $(\mathcal{A}, \mathcal{E})$ is a set $\mathcal{A}$ with a sigma-algebra $\mathcal{E}$ of subsets of $\mathcal{A}$. When we refer to a probability distribution of a random variable taking values in $(\mathcal{A}, \mathcal{E})$ we will work directly with the probability measure on $(\mathcal{A}, \mathcal{E})$ rather than the underlying sample space. For a measurable mapping $T$ from measure space $(\mathcal{A}, \mathcal{E}, \lambda)$ to measurable space $(\mathcal{B}, \mathcal{F})$, we denote by $T_*\lambda$ the push-forward of $\lambda$. $\mathcal{S}^{d-1}$ denotes the unit sphere in $\mathbb{R}^d$ and for any space $\mathcal{A}$, $B(\mathcal{A})$ denotes the Borel $\sigma$-algebra on $\mathcal{A}$. The notation $\mu \ll \nu$ signifies the measure $\mu$ is absolutely continuous with respect to $\nu$. The function clip is defined as $\text{clip}(a, \alpha, \beta) = \min(\beta, \max(\alpha, a))$ for $a \in \mathbb{R}$. If $a \in \mathbb{R}^d$, it is interpreted element-wise.

**Variance of Random Vectors.** We define the variance of a random vector $\boldsymbol{y}$ as $\text{Var}(\boldsymbol{y}) = \mathbb{E}[(\boldsymbol{y} - \mathbb{E}\boldsymbol{y})^\top(\boldsymbol{y} - \mathbb{E}\boldsymbol{y})]$, i.e. the trace of the covariance of y; it is easy to verify standard properties of the variance still hold. This definition is often used to analyze the variance of gradient estimates (Greensmith et al., 2004).

**Stochastic Policy Gradients.** In Section 4 we present marginal policy gradient estimators and work in the very general setting described below. Let $(\mathcal{A}, \mathcal{E}, \mu)$ be a measure space, where as before $\mathcal{A}$ is the action space of the MDP. In practice, we often encounter $(\mathcal{A}, \mathcal{E}) = (\mathbb{R}^d, B(\mathbb{R}^d))$ with $\mu$ as the Lebesgue measure. The types of policies for which there is a meaningful notation of stochastic policy gradients are $\mu$-*compatible measures* (see remarks 2.3 and 2.4).

**Definition 2.1** ($\mu$-Compatible Measures)**.** Let $(\mathcal{A}, \mathcal{E}, \mu)$ be a measure space and consider a parametrized family of measures $\Pi = \{\pi(\cdot, \theta) : \theta \in \Theta\}$ on the same space. $\Pi$ is a $\mu$-compatible family of measures if for all $\theta$:

(a) $\pi(\cdot, \theta) \ll \mu$ with density of the form $f_\pi(\cdot, \theta)$,

(b) $f_\pi$ is differentiable in $\theta$, and

(c) $\pi$ satisfies the conditions to apply the Leibniz integral rule for each $\theta$, so that $\nabla \int_\mathcal{A} f_\pi(a) d\mu = \int_\mathcal{A} \nabla f_\pi(a) d\mu$.

For $\mu$-compatible policies, Theorem 2.2 gives the stochastic policy gradient, easily estimable from samples. When $\mu$ is the counting measure we recover the discrete policy gradient theorem (Sutton

et al., 2000). See Appendix A.1 for a more in depth discussion and a proof of Theorem 2.2, which we include for completeness.

**Theorem 2.2** (Stochastic Policy Gradient). Let $(\mathcal{A}, \mathcal{E}, \mu)$ be a measure space and let $\Pi = \{\pi(\cdot, \theta|s) : \theta \in \Theta\}$ be a family of $\mu$-compatible probability measures. Denoting by $f_\pi$ the density with respect to $\mu$, we have that

$$\nabla \eta = \int_\mathcal{S} d\rho_\pi(s) \int_\mathcal{A} q_\pi(s, a) \nabla \log f_\pi(a|s) d\pi(\cdot|s).$$

In general we want an estimate $g$ of $\nabla \eta$ such that it is unbiased ($\mathbb{E}[g] = \nabla \eta$) and that has minimal variance, so that convergence to a (locally) optimal policy is as fast as possible. In the following sections, we explore a general approach to finding a low variance, unbiased estimator.

**Remark 2.3.** Under certain choices of $T$ (e.g. clipping) the effective action distribution is a mixture of a continuous distribution and point masses. Thus, although it adds some technical overhead, it is necessary that we take a measure theoretic approach in this work.

**Remark 2.4.** Definition 2.1 is required to ensure the policy gradient is well defined, as it stipulates the existence of an appropriate reference measure; it also serves to clarify notation and to draw a distinction between $\pi$ and its density $f_\pi$. Though these details are often minimized they are important in analyzing the interaction between $T$ and $\pi$.

## 3   ANGULAR POLICY GRADIENTS

Consider the task of learning a policy over directions in $\mathcal{A} = \mathbb{R}^2$, or equivalently learning a policy over angles $[0, 2\pi)$. A naive approach is to fit the mean $m_\theta(s)$, model the angle as normally distributed about $m_\theta$, and then clip the sampled angle before execution in the environment. However, this approach is asymmetric in that does not place similar probability on $m_\theta(s) - \epsilon$ and $m_\theta(s) + \epsilon$ for $m_\theta(s)$ near to $0$ and $2\pi$.

An alternative is to model $m_\theta(s) \in \mathbb{R}^2$, sample $a \sim \mathcal{N}(m_\theta(s), \Sigma)$, and then execute $T(a) := a/||a||$ in the environment. This method also works for directional control in $\mathbb{R}^d$. The drawback of this approach is the following: informally speaking, we are sampling from a distribution with $d$ degrees of freedom, but the environment is affected by an action with only $d - 1$ degrees of freedom. This suggests, and indeed we later prove, that the variance of the stochastic policy gradient for this distribution is unnecessarily high. In this section we introduce the angular policy gradient which can be used as a drop-in replacement for the policy update step in existing algorithms.

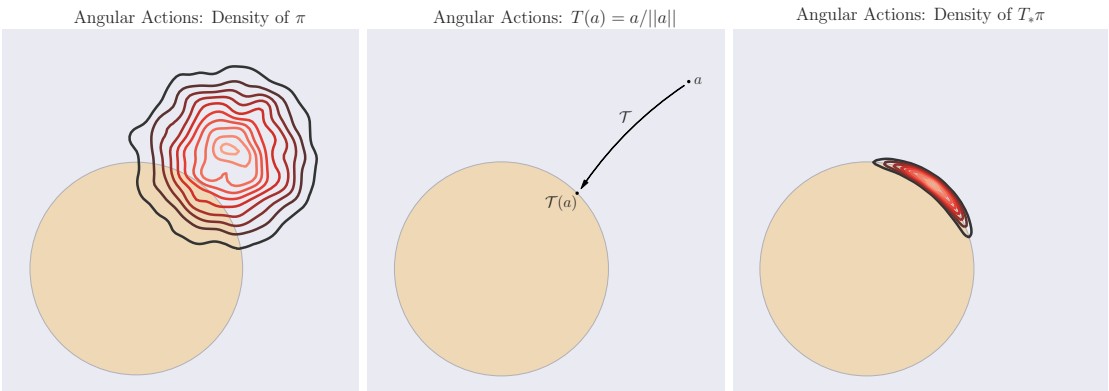

Figure 1: Transformation of a Gaussian policy – (left to right) $\pi(\cdot|s)$, $T = a/||a||$, and $T_*\pi(\cdot|s)$.

### ANGULAR GAUSSIAN DISTRIBUTION

Instead, we can directly model $T(a) \in \mathcal{S}^{d-1}$ instead of $a \in \mathbb{R}^d$. If $a \sim \mathcal{N}(m_\theta(s), \Sigma_\theta(s))$, then $T(a)$ is distributed according to what is known as the angular Gaussian distribution (Definition 3.1). It can be derived by a change of variables to spherical coordinates, followed by integration with respect to

the magnitude of the random vector (Paine et al., 2018). Figure 1 illustrates the transformation of a Gaussian sampling policy $\pi$ under $T$.

**Definition 3.1** (Angular Gaussian Distribution). Let $a \sim \mathcal{N}(m, \Sigma)$. Then, with respect to the spherical measure $\sigma$ on $(\mathcal{S}^{d-1}, B(\mathcal{S}^{d-1}))$, $x = a/||a||$ has density

$$f(x; m, \Sigma) = \left((2\pi)^{d-1}|\Sigma|(x^\top \Sigma^{-1} x)^d\right)^{-1/2} \exp\left(\frac{1}{2}\left(\alpha^2 - m^\top \Sigma^{-1} m\right)\right) \mathcal{M}_{d-1}(\alpha), \quad (3.1)$$

where $\alpha = \frac{x^\top \Sigma^{-1} m}{(x^\top \Sigma^{-1} x)^{1/2}}$ and $\mathcal{M}_{d-1}(x) = (2\pi)^{-\frac{1}{2}} \int_0^\infty u^{d-1} \exp(-(u-x)^2/2) du$.

POLICY GRADIENT METHOD

Although the density in Definition 3.1 does not have a closed form, we can still obtain a stochastic policy gradient for this type of policy. Define the action space as $\mathcal{A} := \mathcal{S}^{d-1}$ and consider angular Gaussian policies parametrized by $\theta := (\theta_m, \theta_\Sigma)$, where $\theta_m$ parametrizes $m$ and $\theta_\Sigma$ parametrizes $\Sigma$. As before, denote the corresponding parametrized family of measures as $\Pi := \{\pi(\cdot, \theta|s) : \theta \in \Theta\}$. Directly from Definition 3.1, we obtain

$$\log f_\pi = \frac{1}{2}\left(\alpha^2 - m^\top \Sigma^{-1} m\right) + \log \mathcal{M}_{d-1}(\alpha) - \frac{1}{2}\left[(d-1)\log 2\pi + \log |\Sigma| + d \log\left(x^\top \Sigma x\right)\right].$$

Though this log-likelihood does not have a closed form, it turns out it is easy to compute the gradient in practice. It is only necessary that we can evaluate $\mathcal{M}'_{d-1}(\alpha)$ and $\mathcal{M}_d(\alpha)$ easily. Assuming for now that we can do so, denote by $\theta_i$ the parameters after $i$ gradient updates and define

$$l_i(\theta) := \frac{1}{2}\left(\alpha^2 - m^\top \Sigma^{-1} m\right) + \underbrace{\frac{\mathcal{M}'_{d-1}(\alpha(\theta_i))}{\mathcal{M}_{d-1}(\alpha(\theta_i))}}_{(i)} \alpha - \frac{1}{2}\left[(d-1)\log 2\pi + \log|\Sigma| + d\log\left(x^\top \Sigma x\right)\right].$$

By design,

$$\nabla \log f_\pi(\theta)|_{\theta=\theta_i} = \nabla l_i(\theta)|_{\theta=\theta_i},$$

thus at update $i$ it suffices to compute the gradient of $l_i$, which can be done using standard auto-differentiation software (Paszke et al., 2017) since term (i) is a constant. From Paine et al. (2018), we have that $\mathcal{M}'_d(\alpha) = d\mathcal{M}_{d-1}(\alpha)$, $\mathcal{M}_{d+1}(\alpha) = \alpha\mathcal{M}_d(\alpha) + d\mathcal{M}_{d-1}(\alpha)$, $\mathcal{M}_1(\alpha) = \alpha\Phi(\alpha) + \phi(\alpha)$ and $\mathcal{M}_0(\alpha) = \Phi(\alpha)$, where $\Phi$, $\phi$ denote the PDF and CDF of $\mathcal{N}(0, 1)$, respectively. Leveraging these properties, the integral $\mathcal{M}_d(\alpha)$ can be computed recursively; Algorithm 1 in Appendix B.1 gives psuedo-code for the computation. Importantly it runs in $\mathcal{O}(d)$ time and therefore does not effect the computational cost of the policy update since it is dominated by the cost of computing $\nabla l_i$. In addition, stochastic gradients of policy loss functions for TRPO or PPO Schulman et al. (2015; 2017) can be computed in a similar way since we can easily get the derivative of $f_\pi(\theta)$ when $\mathcal{M}_{d-1}(\alpha)$ and $\mathcal{M}'_{d-1}(\alpha)$ are known.

## 4 MARGINAL POLICY GRADIENT ESTIMATORS

In Section 2, we described a general setting in which a stochastic policy gradient theorem holds on a measure space $(\mathcal{A}, \mathcal{E}, \lambda)$ for a family of $\lambda$-compatible probability measures, $\Pi = \{\pi(\cdot, \theta|s) : \theta \in \Theta\}$. As before, we are interested in the case when the dynamics of the environment only depend on $a \in \mathcal{A}$ via a function $T$. That is to say $r(s, a) := r(s, T(a))$ and $p(s, a, s') := p(s, T(a), s')$.

The key idea in Marginal Policy Gradient is to replace the policy gradient estimate based on the log-likelihood of $\pi$ with a lower variance estimate, which is based on the log-likelihood of $T_*\pi$. $T_*\pi$ can be thought of as (and in some cases is) a marginal distribution, hence the name *Marginal Policy Gradient*. For this reason it can easily be used with value function approximation and GAE, as well as incorporated into algorithms like TRPO, A3C and PPO.

### 4.1 SETUP AND REGULARITY CONDITIONS

For our main results we need regularity Condition 4.1 on the measure space $(\mathcal{A}, \mathcal{E}, \lambda)$. Next, let $(\mathcal{B}, \mathcal{F})$ be another measurable space and $T : \mathcal{A} \to \mathcal{B}$ be a measurable mapping. $T$ induces a family of

probability measures on $(\mathcal{B}, \mathcal{F})$, denoted $T_*\Pi := \{T_*\pi(\cdot, \theta|s) : \theta \in \Theta\}$. We also require regularity Conditions 4.2 and 4.3 regarding the structure of $\mathcal{F}$ and the existence of a suitable reference measure $\mu$ on $(\mathcal{B}, \mathcal{F})$. These conditions are all quite mild and are satisfied in all practical settings, to the best of our knowledge.

**Condition 4.1** . $\mathcal{A}$ is a metric space and $\lambda$ is a Radon measure.[1]

**Condition 4.2** . $\mathcal{F}$ is countably generated and contains the singleton sets $\{b\}$, for all $b \in \mathcal{B}$.

**Condition 4.3** . There exists a $\sigma$-finite measure $\mu$ on $(\mathcal{B}, \mathcal{F})$ such that $T_*\lambda \ll \mu$ and $T_*\Pi$ is $\mu$-compatible.

In statistics, Fisher information is used to capture the variance of a score function. In reinforcement learning, typically one encounters a score function that has been rescaled by a measurable function $q(a)$. Definition 4.4 provides a variant of Fisher information for $\lambda$-compatible distributions and rescaled score functions; we defer a discussion of the definition until Section 4.4 after we present our results in their entirety. If $q(a) = 1$, Definition 4.4 is the trace of the classical Fisher Information.

**Definition 4.4** (Total Scaled Fisher Information)**.** Let $(\mathcal{A}, \mathcal{E}, \lambda)$ be a measure space, $\Pi = \{\pi(\cdot, \theta) : \theta \in \Theta\}$ be a family of $\lambda$-compatible probability measures, and $q$ a measurable function on $\mathcal{E}$. The total scaled fisher information is defined as $\mathcal{I}_{\pi,\lambda}(q, \theta) := \mathbb{E}[q(a)^2 \nabla \log f_\pi(a)^\top \nabla \log f_\pi(a)]$.

## 4.2 Variance Reduction Guarantee

From Theorem 2.2 it is immediate that

$$\nabla \eta(\boldsymbol{\theta}) = \int_\mathcal{S} d\rho(s) \int_\mathcal{A} q(T(a), s) \nabla \log f_\pi(a|s) d\pi(a|s)$$
$$= \int_\mathcal{S} d\rho(s) \int_\mathcal{B} q(b, s) \nabla \log f_{T_*\pi}(b|s) d(T_*\pi)(b|s),$$

where we dropped the subscripts on $\rho$ and $q$ because the two polices affect the environment in the same way, and thus have the same value function and discounted state occupancy measure. Denote the two alternative gradient estimators as $g_1 = q(T(a), s) \nabla \log f_\pi(a|s)$ and $g_2 = q(b, s) \nabla \log f_{T_*\pi}(b|s)$. Just by definition, we have that $\mathbb{E}_{\rho,\pi}[g_1] = \mathbb{E}_{\rho,\pi}[g_2]$. Lemma 4.5 says something slightly different – it says that they are also equivalent in expectation conditional on the state $s$, a fact we use later.

**Lemma 4.5.** Let $(\mathcal{A}, \mathcal{E}, \lambda)$ and $(\mathcal{B}, \mathcal{F}, \mu)$ be measure spaces, and $T : \mathcal{A} \to \mathcal{B}$ be measurable mapping. If $\Pi$, parametrized by $\theta$, is $\lambda$-compatible and $T_*\Pi$ is $\mu$-compatible, then

$$\mathbb{E}_{\pi|s}[g_1] = \mathbb{E}_{\pi|s}[g_2] = \mathbb{E}_{T_*\pi|s}[g_2]. \tag{4.1}$$

*Proof.* The result follows immediately from the proof of Theorem 2.2 in Appendix A.1. $\square$

Because the two estimates $g_1$ and $g_2$ are both unbiased, it is always preferable to use whichever has lower variance. Theorem 4.6 shows that $g_2$ is the lower variance policy gradient estimate. See Appendix B.3 for the proof. The implication of Theorem 4.6 is that if there is some information loss via a function $T$ before the action interacts with the dynamics of the environment, then one obtains a lower variance estimator of the gradient by replacing the density of $\pi$ with the density of $T_*\pi$ in the expression for the policy gradient.

**Theorem 4.6.** Let $g_1$ and $g_2$ be as defined above. Then if Conditions 4.1-4.3 are satisfied,

$$\mathrm{Var}_{\rho,\pi}(g_1) - \mathrm{Var}_{\rho,T_*\pi}(g_2) = \mathbb{E}_{\rho,T_*\pi} \left[ \mathcal{I}_{\pi|s|b,\lambda_b}(q \circ T, \theta) \right] \geq 0,$$

for some family of measures $\{\lambda_b\}$ on $\mathcal{A}$.

## 4.3 Examples of Marginal Policy Gradient Estimators

### Clipped Action Policy Gradient

Consider a control problem where actions in $\mathbb{R}$ are clipped to an interval $[\alpha, \beta]$. Let $\lambda$ be an arbitrary measure on $(\mathcal{A}, \mathcal{E}) := (\mathbb{R}, B(\mathbb{R}))$, and consider any $\lambda$-compatible family $\Pi$. Following Fujita &

---

[1]On a metric space $\mathcal{A}$, a Radon measure is a measure defined on the Borel $\sigma$-algebra for which each compact $K \subset \mathcal{A}$, $\lambda(K) < \infty$ and for all $B \in B(\mathcal{A})$, $\lambda(B) = \sup_{K \subseteq B} \lambda(K)$ where $K$ is compact.

Maeda (2018), define the clipped score function

$$\widetilde{\psi}(s,b,\theta) = \begin{cases} \nabla \log \int_{(-\infty,\alpha]} f_\pi(a,\theta|s)d\lambda & b = \alpha \\ \nabla \log f_\pi(b,\theta|s) & b \in (\alpha,\beta) \\ \nabla \log \int_{[\beta,\infty)} f_\pi(a,\theta|s)d\lambda & b = \beta. \end{cases}$$

We can apply Theorem 4.6 in this setting to obtain Corollary 4.7. It is a strict generalization of the results in Fujita & Maeda (2018) in that it applies to a larger class of measures and provides a much stronger variance reduction guarantee. It is possible to obtain this more powerful result precisely because we require minimal assumptions for Theorem 4.6. Note that the result can be extended to $\mathbb{R}^d$, but we stick to $\mathbb{R}$ for clarity of presentation. See Appendix B.4 for a discussion of which distributions are $\lambda$-compatible and a proof of Corollary 4.7.

**Corollary 4.7.** Let $\lambda$ be an arbitrary measure on $(\mathcal{A},\mathcal{E}) := (\mathbb{R}, B(\mathbb{R}))$, $T(a) := \text{clip}(a,\alpha,\beta)$, and $\psi(s,a,\theta) := \nabla \log f_\pi(a,\theta|s)$. If $\Pi$ is a $\lambda$-compatible family parametrized by $\theta$ and the dynamics of the environment depend only on $T(a)$, then

1. $\mathbb{E}_{\pi|s}[q_\pi(s,a)\psi(s,a,\theta)] = \mathbb{E}_{\pi|s}\left[q_\pi(s,a)\widetilde{\psi}(s,T(a),\theta)\right]$, and

2. $\text{Var}_{\rho,\pi}(q_\pi(s,a)\psi(s,a,\theta)) - \text{Var}_{\rho,\pi}(q_\pi(s,a)\widetilde{\psi}(s,T(a),\theta)) = \mathbb{E}_\rho\left[\mathbb{E}_{T_*\pi|s}\left[\mathcal{I}_{\pi|s|b,\lambda_b}(q\circ T,\theta)\right]\right]$, for some family of measures $\{\lambda_b\}$ on $\mathcal{A}$.

### ANGULAR POLICY GRADIENT

Now consider the case where we sample an action $a \in \mathbb{R}^d$ and apply $T(a) = a/||a||$ to map into $\mathcal{S}^{d-1}$. Let $(\mathcal{A},\mathcal{E}) = (\mathbb{R}^d, B(\mathbb{R}^d))$ and let $\lambda$ be the Lebesgue measure. When $\Pi$ is a multivariate Gaussian family parametrized by $\theta$, $T_*\Pi$ is an angular Gaussian family also parametrized by $\theta$ (Section 3). If $\Pi$ is $\lambda$-compatible – here it reduces to ensuring the parametrization is such that $f_\pi$ is differentiable in $\theta$ – then $T_*\Pi$ is $\sigma$-compatible, where $\sigma$ denotes the spherical measure. Denoting by $f_{MV}(a,\theta|s)$ and $f_{AG}(b,\theta|s)$ the corresponding multivariate and angular Gaussian densities, respectively, we state the results for this setting as Corollary 4.8. See Appendix B.4 for a proof.

**Corollary 4.8.** Let $\lambda$ be the Lebesgue measure on $(\mathcal{A},\mathcal{E}) = (\mathbb{R}^d, B(\mathbb{R}^d))$, $T(a) := a/||a||$ and $\Pi$ be a multivariate Gaussian family on $\mathcal{A}$ parametrized by $\theta$. If the dynamics of the environment only depend on $T(a)$ and $f_{MV}(\cdot,\theta|s)$, the density corresponding to $\Pi$, is differentiable in $\theta$, then

1. $\mathbb{E}_{\pi|s}[q_\pi(s,a)\psi(s,a,\theta)] = \mathbb{E}_{\pi|s}\left[q_\pi(s,a)\widetilde{\psi}(s,T(a),\theta)\right]$, and

2. $\text{Var}_{\rho,\pi}(q_\pi(s,a)\psi(s,a,\theta)) - \text{Var}_{\rho,\pi}(q_\pi(s,a)\widetilde{\psi}(s,T(a),\theta)) = \mathbb{E}_{\rho,T_*\pi}\left[\text{Var}_{\pi|b}(q_\pi(s,a)\psi_r(s,r,\theta))\right]$,

where $r = ||a||$, $f_r$ is the conditional density of $r$, $\psi(s,a,\theta) := \nabla \log f_{MV}(a,\theta|s)$, $\widetilde{\psi}(s,b,\theta) = \nabla \log f_{AG}(b,\theta|s)$, and $\psi_r(s,r,\theta) = \nabla \log f_r(r,\theta|s)$.

### PARAMETRIZED ACTION SPACES

As one might expect, our variance reduction result applies to parametrized action spaces when a lossy transformation $T_i$ is applied to the parameter for discrete action $i$. See Appendix B.5 for an in depth discussion of policy gradient methods for parametrized action spaces.

## 4.4 DISCUSSION

Denoting by $g_1$ the standard policy gradient estimator for a $\lambda$-compatible family $\Pi$, observe that $\text{Var}_{\rho,\pi}(g_1) = \mathcal{I}_{\pi,\lambda}(q,\theta)$. We introduce the quantity $\mathcal{I}_{\pi,\lambda}$ because unless $T$ is a coordinate projection it is not straightforward to write Theorem 4.6 in terms of the density of a conditional distribution. Corollary 4.8 can be written this way because under a re-parametrization to polar coordinates, $T(a) = a/||a||$ can be written as a coordinate projection. In general, by using $\mathcal{I}_{\pi,\lambda}$ we can phrase the result in terms of a quantity with an intuitive interpretation: a ($q$-weighted) measure of information contained in $a$ that *does not influence* the environment.

Recalling the law of total variance (LOTV), we can observe that Theorem 4.6 is indeed specific version of that general result. We can not directly apply the LOTV because in the general setting, it is highly non-trivial to conclude that $g_2$ is a version of the conditional expectation of $g_1$, and for arbitrary policies, one must be extremely careful when making the conditioning argument (Chang

& Pollard, 1997). However for certain special cases, like CAPG, we can check fairly easily that $g_2 = \mathbb{E}[g_1|b]$.

# 5 APPLICATIONS AND DISCUSSION

## 5.1 2D NAVIGATION TASK

Because relatively few existing reinforcement learning environments support angular actions, we implement a navigation task to benchmark our methods[2]. In this navigation task, the agent is located on a platform and must navigate from one location to another without falling off. The state space is $\mathcal{S} = \mathbb{R}^2$, the action space is $\mathcal{A} = \mathbb{R}^2$ and the transformation $T(a) = a/||a||$ is applied to actions before execution in the environment. Let $s_G = (1, 1)$ be the goal (terminal) state. Using the *reward shaping* approach (Ng et al., 1999), we define a potential function $\phi(s) = ||s - s_G||_2$ and a reward function as $r(s_t, a_t) = \phi(s_t) - \phi(s_t + a_t)$. The start state is fixed at $s_0 = (-1, -1)$. One corner of the platform is located at $(-1.5, -1.5)$ and the other at $(1.5, 1.5)$.

We compare angular Gaussian policies with (1) bivariate Gaussian policies and (2) a 1-dimensional Gaussian policy where we model the mean of the angle directly, treating angles that differ by $2\pi$ as identical. For all candidate policies, we use A2C (the synchronous version of A3C (Mnih et al., 2016)) to learn the conditional mean $m(s; \theta)$ of the sampling distribution by fitting a feed-forward neural network with tanh activations. The variance of the sampling distribution, $\sigma^2\mathbf{I}$, is fixed. For the critic we estimate the state value function $v_\pi(s)$, again using a feed-forward neural network. Appendix C.1 for details on the hyper-parameter settings, network architecture and training procedure.

## 5.2 APPLICATION – KING OF GLORY

We implement a marginal policy gradient method for *King of Glory* (the North American release is titled *Arena of Valor*) by Tencent Games. *King of Glory* has several game types and we focus on the 1v1 version. Our work here is one of the first attempts to solve *King of Glory*, and MOBA games in general, using reinforcement learning. Similar MOBA games include Dota 2 and League of Legends.

**Game Description.**  In *King of Glory*, players are divided into two "camps" located in opposite corners of the game map. Each player chooses a "hero", a character with unique abilities, and the objective is to destroy the opposing team's "crystal", located at their game camp. The path to each camp and crystal is guarded by towers which attack enemies when in range. Each team has a number of allied "minions", less powerful characters, to help them destroy the enemy crystal. Only the "hero" is controlled by the player. During game play, heroes increase in level and obtain gold by killing enemies. This allows the player to upgrade the level of their hero's unique skills and buy improved equipment, resulting in more powerful attacks, increased HP, and other benefits. Figure 2 shows *King of Glory* game play; in the game pictured, both players use the hero "Di Ren Jie".

**Formulation as an MDP.**  $\mathcal{A}$ is a parametrized action space with 7 discrete actions, 4 of which are parametrized by $\omega \in \mathbb{R}^2$. These actions include move, attack, and use skills; a detailed description of all actions and parameters is given in Table 3, Appendix C.2. In our setup, we use rules crafted by domain experts to manage purchasing equipment and learning skills. The transformation $T(a) = a/||a||$ is applied to the action parameter before execution in the environment, so the effective action parameter spaces are $\mathcal{S}^1$.

Using information obtained directly from the game engine, we construct a 2701-dimensional state representation. Features extracted from the game engine include hero locations, hero health, tower health, skill availability and relative locations to towers and crystals – see Appendix C.2 for details on the feature extraction process. As in Section 5.1, we define rewards using a potential function. In particular we define a reward feature mapping $\rho$ and a weighting vector $w$, and then a linear potential function as $\phi_r(s) = w^T\rho(s)$. Information extracted by $\rho$ includes hero health, crystal health, and game outcome; see Table 5, Appendix C.2 for a complete description of $w$ and $\rho$. Using $\phi_r$, we can define the reward as $r_t = \phi_r(s_t) - \phi_r(s_{t-1})$.

---

[2]We have made this environment and the implementation used for the experiments available on-line. We temporarily removed the link from this paper to preserve anonymity.

**Implementation.** We implement the A3C algorithm, and model both the policy $\pi$ and the value function $v_\pi$ using feed-forward neural networks. See Appendix C.2 for more details on how we model and learn the value function and policy. Using the setup described above, we compare:

1. a standard policy gradient approach for parametrized action spaces, and
2. a marginal (angular) policy gradient approach, adapted to the parametrized action space where $T_i(a) = a/||a||$ is applied to parameter $i$.

Additional details on both approaches can be found in Appendix B.5.

## 5.3 RESULTS

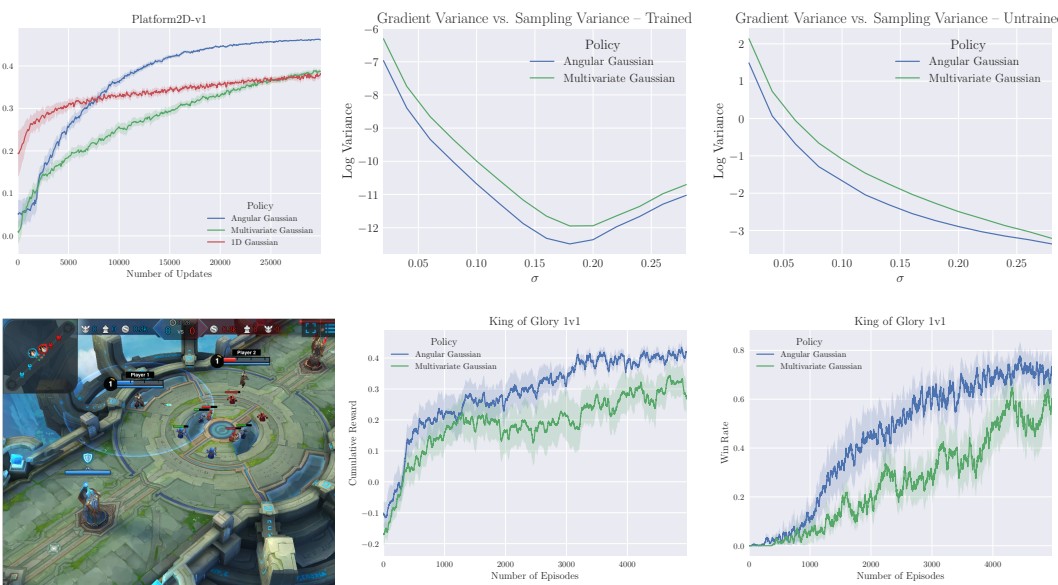

Figure 2: On top are results for Platform2D-v1; on bottom, results for King of Glory 1v1 and a screenshot of game play.

For the navigation task, the top row of Figure 2 contains, from left to right, cumulative, discounted reward trajectories, and two plots showing the variances of the competing estimators. We see that the agent using the angular policy gradient converges faster compared to the multivariate Gaussian due to the variance reduced gradient estimates. The second baseline also performs worse than APG, likely due in part to the fact that the critic must approximate a periodic function. Only APG achieves the maximum possible cumulative, discounted reward. On the *King of Glory* 1 vs. 1 task, the agent is trained to play as the hero Di Ren Jie and training occurs by competing with the game's internal AI, also playing as Di Ren Jie. The bottom row of Figure 2 shows the results, and as before, the angular policy gradient outperforms the standard policy gradient by a significant margin both in terms of win percentage and cumulative discounted reward.

In addition, Figure 2 highlights the effects of Theorem 4.6 in practice. The plot in the center shows the variance at the start of training, for a fixed random initialization, and the plot on the right shows the variance for a trained model that converged to the optimal policy. The main difference between the two settings is that the value function estimate $\widehat{v}_\pi$ is highly accurate for the trained model (since both actor and critic have converged) and highly inaccurate for the untrained model. In both cases, we see that the variance of the marginal policy gradient estimator is roughly $\frac{1}{2}$ that of the estimator using the sampling distribution.

## 5.4 DISCUSSION

Motivated by challenges found in complex control problems, we introduced a general family of variance reduced policy gradients estimators. This view provides the first unified approach to

problems where the environment only depends on the action through some transformation $T$, and we demonstrate that CAPG and APG are members of this family corresponding to different choices of $T$. We also show that it can be applied to parametrized action spaces. Because thorough experimental work has already been done for the CAPG member of the family (Fujita & Maeda, 2018), confirming the benefits of MPG estimators, we do not reproduce those results here. Instead we focus on the case when $T(a) = a/||a||$ and demonstrate the effectiveness of the angular policy gradient approach on *King of Glory* and our own Platform2D-v1 environment. Although at this time few RL environments use directional actions, we anticipate the number will grow as RL is applied to newer and increasingly complex tasks like MOBA games where such action spaces are common. We also envision that our methods can be applied to autonomous vehicle, in particular quadcopter, control.

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

# A    ADDITIONAL PRELIMINARIES

This section contains additional preliminary material and discussion thereof.

## A.1    DISCUSSION – STOCHASTIC POLICY GRADIENTS

We require a stochastic policy gradient theorem that can be applied to distributions on arbitrary measurable spaces in order to rigorously analyze the Marginal Policy Gradients framework. Let the notation be as in Section 2. The first ingredient is Proposition A.1, which gives a very general form of policy gradient, defined for an arbitrary probability measure.

**Proposition A.1.** [Ciosek & Whiteson (2018)] Let $\pi(\cdot|s)$ be a probability measure on $(\mathcal{A}, \mathcal{E})$, then

$$\nabla\eta = \int_{\mathcal{S}} d\rho_\pi(s) \left[\nabla v_\pi(s) - \int_{\mathcal{A}} d\pi(a|s)\nabla q_\pi(s,a)\right].$$

This is an important step towards the form of stochastic policy gradient theorem we need in order to present our unified analysis that includes measures with uncountable support and also those which do not admit a density with respect to Lebesgue measure – something frequently encountered in practice. To obtain a stochastic policy gradient theorem from Proposition A.1 we simply need to replace $\nabla v_\pi(s)$ with an appropriate expression. As in Ciosek & Whiteson (2018), we need to be able to justify an interchange along the lines of

$$\nabla v_\pi = \nabla \int_{\mathcal{A}} d\pi(a|s)q_\pi(s,a) = \int_{\mathcal{A}} da\nabla\pi(a|s)q_\pi(s,a) + \int_{\mathcal{A}} d\pi(a|s)\nabla q_\pi(s,a). \tag{A.1}$$

Such an expression doesn't make sense for arbitrary $\pi$, so we must be precise regarding the conditions under which such an expression makes sense and the interchange is permitted, hence Definition 2.1. Because we did not find a statement with the sort of generality we required in the literature, we give a proof of our statement the stochastic policy gradient theorem, Theorem 2.2, below.

*Proof of Theorem 2.2.* The proof follows standard arguments. Because $\Pi$ is $\mu$-compatible we obtain that

$$\begin{aligned}
\nabla v_\pi &= \nabla \int_{\mathcal{A}} d\pi(a|s)q_\pi(s,a) \\
&= \int_{\mathcal{A}} \nabla\left[q_\pi(s,a)f_\pi(a|s)\right] d\mu \\
&= \int_{\mathcal{A}} q_\pi(s,a)\nabla f_\pi(a|s)d\mu + \int_{\mathcal{A}} \nabla q_\pi(s,a)d\pi(a|s).
\end{aligned}$$

The result now follows immediately from Proposition A.1. $\qquad\square$

## A.2    DISINTEGRATION THEOREMS

The definitions and propositions below are from Chang & Pollard (1997), which we include here for completeness. Let $(\mathcal{A}, \mathcal{E}, \lambda)$ be a measure space and $(\mathcal{B}, \mathcal{F})$ a measurable space. Let $\lambda$ be a $\sigma$-finite measure on $\mathcal{E}$ and $\mu$ be a $\sigma$-finite measure on $\mathcal{F}$.

**Definition A.2** (($T, \mu$)-disintegration, Chang & Pollard (1997))**.** The measure $\lambda$ has a $(T, \mu)$-disintegration, denoted $\{\lambda_b\}$ if for all nonnegative measurable $f$ on $\mathcal{A}$

- $\lambda_b$ is a $\sigma$-finite measure on $\mathcal{E}$ that is concentrated on $E_b := \{T = b\}$ in the sense that $\int_{\mathcal{A}} \mathbb{I}[\mathcal{A} \setminus E_b]d\lambda_b = 0$ for $\mu$-almost all $b$,

- the function $b \to \int_{T^{-1}(b)} f d\lambda_b$ is measurable, and

- $\int_{\mathcal{A}} f d\lambda = \int_{\mathcal{B}} \int_{T^{-1}(b)} f d\lambda_b d\mu$.

If $\mu = T_*\lambda$, then we call $\lambda_b$ a $T$-disintegration. With some additional assumptions, we have the existence theorem given below.

**Proposition A.3** (Existence, Chang & Pollard (1997))**.** Let $\mathcal{A}$ be a metric space, $\lambda$ be a $\sigma$-finite Radon measure, and $\mu$ be a $\sigma$-finite measure such that $T_*\lambda \ll \mu$. If $\mathcal{F}$ is countably generated and contains the singleton sets $\{b\}$, then $\lambda$ has a $(T, \mu)$-disintegration. The measures $\{\lambda_b\}$ are unique up to an almost-sure equivalence in that if $\{\lambda_b^*\}$ is another $(T, \mu)$-disintegration, $\mu(\{b : \lambda_b \neq \lambda_b^*\}) = 0$.

Lastly, we have Proposition A.4 which characterizes the properties of disintegrations and how they relate to densities and push-forward measures.

**Proposition A.4** (Chang & Pollard (1997))**.** Let $\lambda$ have a $(T, \mu)$-disintegration $\{\lambda_b\}$, and let $\rho$ be absolutely continuous with respect to $\lambda$ with a finite density $r(a)$, where each of $\lambda$, $\mu$ and $\rho$ is $\sigma$-finite. Then

- $\rho$ has a $(T, \mu)$-disintegration $\{\rho_b\}$ where $\rho_b \ll \lambda_b$ with density $r(a)$,

- $T_*\rho \ll \mu$ with density $r_T(b) := \int_{T^{-1}(b)} r(a)d\lambda_b$,

- the measures $\{\rho_b\}$ are finite for $\mu$ almost all $b$ if and only if $T_*\rho$ is $\sigma$-finite,

- the measures $\{\rho_b\}$ are probabilities for $\mu$ almost all $b$ if and only if $\mu = T_*\rho$, and

- if $T_*\rho$ is $\sigma$-finite, then $T_*\rho(\{b : r_T(b) = 0\}) = 0$ and $T_*\rho(\{b : r_T(b) = \infty\}) = 0$. For $T_*\rho$-almost all $b$, the measures $\{\widetilde{\rho}_b\}$ defined by

$$\int_{T^{-1}(b)} f(a)d\widetilde{\rho}_b = \int_{T^{-1}(b)} f(a)r_{a|b}(a)d\lambda_b \ \text{ and } \ r_{a|b}(a) := \mathbb{I}[0 < r_T(b) < \infty]\frac{r(a)}{r_T(b)},$$

  are probability measures that give a $T$-disintegration of $\rho$.

# B    Theory and Methodology

This section contains additional theoretical and methodology results, including our crucial scaled Fisher information decomposition theorem.

## B.1    Angular Policy Gradient

Algorithm 1 shows how to compute $\mathcal{M}_d(\alpha)$, allowing us to easily find the angular policy gradient.

---

**Algorithm 1** Computing $\mathcal{M}_d(\alpha)$ for Angular Policy Gradient

---

**Input:** $d, \alpha$
**Output:** $\mathcal{M}_d(\alpha)$
 1: $M_0 \leftarrow \Phi(\alpha)$
 2: $M_1 \leftarrow \alpha\Phi(\alpha) + \phi(\alpha)$
 3: **if** $d > 1$ **then**
 4:     **for** $i = 2, \ldots, d$ **do**
 5:         $M_i \leftarrow \alpha M_{i-1} + dM_{i-2}$
 6:     **end for**
 7: **end if**
 8: **return** $M_d$

---

## B.2    Fisher Information Decomposition

Using the disintegration results stated in Appendix A.2, we now can state and prove our key decomposition result, Theorem B.1, used in the proof of our main result.

**Theorem B.1** (Fisher Information Decomposition). Let $(\mathcal{A}, \mathcal{E}, \lambda)$ be a measure space, $(\mathcal{B}, \mathcal{F})$ be a measurable space, $T : \mathcal{A} \to \mathcal{B}$ be a measurable, surjective mapping, and $q$ a measurable function on $\mathcal{F}$. Consider a $\lambda$-compatible family of probability measures $\Pi = \{\pi(\cdot, \theta) : \theta \in \Theta\}$ on $\mathcal{E}$ and denote $T_*\Pi := \{T_*\pi(\cdot, \theta) : \theta \in \Theta\}$, a family of measures on $\mathcal{F}$. If

    (a)  $\mathcal{A}$ is a metric space, $\lambda$ is a Radon measure, and $T_*\lambda \ll \mu$ for a $\sigma$-finite measure $\mu$ on $\mathcal{F}$;

    (b)  $\mathcal{F}$ is countably generated and contains the singleton sets $\{b\}$;

    (c)  $T_*\Pi$ is a $\mu$-compatible family for a measure $\mu$ on $\mathcal{F}$;

then

    1.  $\lambda$ has a $(T, \mu)$-disintegration $\{\lambda_b\}$;

    2.  $\Pi|b$ is a $\lambda_b$-compatible family of probability measures that give a $T$-disintegration of $\pi$;

    3.  for any measurable function $q : \mathcal{B} \to \mathbb{R}$,

$$\mathcal{I}_{\pi,\lambda}(q \circ T, \theta) = \mathbb{E}_{T_*\pi}\left[\mathcal{I}_{\pi|b,\lambda_b}(q \circ T, \theta)\right] + \mathcal{I}_{T_*\pi,\mu}(q, \theta).$$

*Proof of Theorem B.1.* To simplify matters, we assume without loss of generality that all densities are strictly positive. This is allowed because if some density is zero on part of its domain, we can just replace the associated measure with its restriction to sets where the density is non-zero.

The conditions to apply Proposition A.3 are satisfied, so $\lambda$ has a $(T, \mu)$-disintegration $\{\lambda_b\}$, which proves claim 1. Next, denote by $g(a) = \nabla \log f_\pi(a)$ and $h(b) = \nabla \log f_{T_*\pi}(b)$. Because the

conditions to apply Proposition A.4 are satisfied, we obtain that

$$
\begin{aligned}
\int_{\mathcal{A}} q(T(a))^2 g(a)^\top g(a) d\pi(a) &= \int_{\mathcal{B}} \int_{T^{-1}(b)} q(T(a))^2 g(a)^\top g(a) f_{a|b}(a) d\lambda_b(a) dT_*\pi(b) \\
&= \int_{\mathcal{B}} q(b)^2 \int_{T^{-1}(b)} g(a)^\top g(a) f_{a|b}(a) d\lambda_b(a) dT_*\pi(b) \\
&= \int_{\mathcal{B}} q(b)^2 \int_{T^{-1}(b)} \left[ \nabla \log f_\pi(a)^\top \nabla \log f_\pi(a) \right] f_{a|b}(a) d\lambda_b(a) dT_*\pi(b).
\end{aligned}
$$
$$(B.1)$$

Denoting by $\pi|b$ the probability measure with density $f_{a|b}$, we see that $\Pi|b := \{\pi|b(\cdot,\theta) : \theta \in \Theta\}$ is a $\lambda_b$-compatible family of probability measures, proving claim 2.

If we denote $\mathbb{E}_{\pi|b}[g] = \int_{T^{-1}(b)} g(a) f_{a|b}(a) d\lambda_b(a)$, we further obtain that

$$
\begin{aligned}
(B.1) &= \int_{\mathcal{B}} q(b)^2 \mathbb{E}_{\pi|b}[\nabla \log f_{a|b}(a)^\top \nabla \log f_{a|b}(a)] dT_*\pi(b) \\
&+ \underbrace{2 \int_{\mathcal{B}} q(b)^2 \nabla \log f_T(b)^\top \mathbb{E}_{\pi|b}[\nabla \log f_{a|b}(a)] dT_*\pi(b)}_{(i)} \\
&+ \int_{\mathcal{B}} q(b)^2 \mathbb{E}_{\pi|b}[\nabla \log f_T(b)^\top \nabla \log f_T(b)] dT_*\pi(b).
\end{aligned}
$$

In the equation above, term (i) is 0 because $\mathbb{E}_{\pi|b}[\nabla \log f_{a|b}(a)] = 0$. Thus we get that

$$
\begin{aligned}
(B.1) &= \int_{\mathcal{B}} q(b)^2 \mathbb{E}_{\pi|b}[\nabla \log f_{a|b}(a)^\top \nabla \log f_{a|b}(a)] dT_*\pi(b) + \int_{\mathcal{B}} q(b)^2 \nabla \log f_T(b)^\top \nabla \log f_T(b) dT_*\pi(b) \\
&= \int_{\mathcal{A}} q(T(a))^2 \nabla \log f_{a|b}(a)^\top \nabla \log f_{a|b}(a) d\pi(a) + \int_{\mathcal{B}} q(b)^2 \nabla \log f_T(b)^\top \nabla \log f_T(b) dT_*\pi(b).
\end{aligned}
$$
$$(B.2)$$

Because a density is unique almost-everywhere, we can replace $f_T$ with $f_{T_*\pi}$ in (B.2), giving claim 3:

$$
\mathbb{E}_\pi \left[ q(T(a))^2 g(a)^\top g(a) \right] = \mathbb{E}_{T_*\pi} \left[ q(b)^2 \mathbb{E}_{\pi|b} \left[ \nabla \log f_{a|b}(a)^\top \nabla \log f_{a|b}(a) \right] \right] + \mathbb{E}_{T_*\pi} \left[ q(b)^2 h(b)^\top h(b) \right]
$$

$$\Updownarrow$$

$$
\mathcal{I}_{\pi,\lambda}(q \circ T, \theta) = \mathbb{E}_{T_*\pi} \left[ \mathcal{I}_{\pi|b,\lambda_b}(q \circ T, \theta) \right] + \mathcal{I}_{T_*\pi,\mu}(q, \theta).
$$

$\square$

### B.3 PROOF OF THEOREM 4.6

First, we decompose the variance of $g_1$ as

$$
\operatorname{Var}_{\rho,\pi}(g_1) = \operatorname{Var}_\rho \left[ \mathbb{E}_{\pi|s}[g_1] \right] + \mathbb{E}_\rho \left[ \operatorname{Var}_{\pi|s}[g_1] \right].
$$
$$(B.3)$$

A similar decomposition holds for $g_2$. By combining Lemma 4.5 with (B.3) and its equivalent for $g_2$, we get that

$$
\operatorname{Var}_{\rho,\pi}(g_1) - \operatorname{Var}_{\rho,T_*\pi}(g_2) = \mathbb{E}_\rho \left[ \operatorname{Var}_{\pi|s}[g_1] - \operatorname{Var}_{T_*\pi|s}[g_2] \right].
$$

For any fixed $s$, applying the definition of variance given in Section 2 and Lemma 4.5 gives

$$
\operatorname{Var}_{\pi|s}[g_1] - \operatorname{Var}_{\pi|s}[g_2] = \mathbb{E}_{\pi|s} \left[ g_1^\top g_1 - g_2^\top g_2 \right].
$$
$$(B.4)$$

By applying Theorem B.1 (see Appendix B.2), to $\mathbb{E}_{\pi|s} \left[ g_1^\top g_1 \right]$ we obtain

$$
\mathbb{E}_{\pi|s} \left[ g_1^\top g_1 - g_2^\top g_2 \right] = \mathbb{E}_{T_*\pi} \left[ \mathcal{I}_{\pi|b,\lambda_b}(q \circ T, \theta) \right].
$$
$$(B.5)$$

The result follows from combining (B.4) and (B.5), concluding the proof.

### B.4 Marginal Policy Gradients for Clipped and Normalized Actions

For the clipped action setting, we give an example of a $\lambda$-compatible family for which Corollary 4.7 can be applied.

**Example B.2** (The Gaussian is $\lambda$-compatible)**.** Let $(\mathcal{A}, \mathcal{E}) := (\mathbb{R}, B(\mathbb{R}))$ and $\lambda$ be the Lebesgue measure. Consider $\Pi$, a Gaussian family parametrized by $\theta \in \Theta$. If $\Theta$ is constrained such that the variance is lower bounded by $\epsilon > 0$, $\Pi$ is $\lambda$-compatible.

Below are proofs of Corollaries 4.7 and 4.8 from Section 4.

*Proof of Corollary 4.7.* First, it is clear $T$ is measurable, and it is easy to confirm that Conditions 4.1-4.3 hold. Next, define $\mu = \delta_\alpha + \delta_\beta + \lambda$, where $\lambda$ is understood to be its restriction to $(\alpha, \beta)$. As defined, $\mu$ is a mixture measure on $\mathcal{B}$ and we can easily check that $T_*\Pi$ is $\mu$-compatible. In fact, the density of $T_*\pi(\cdot, \theta|s)$ is given by

$$f_{T_*\pi}(b, \theta) = \begin{cases} \int_{(-\infty, \alpha]} f_\pi(a, \theta)d\lambda & b = \alpha \\ f_\pi(b, \theta) & b \in (\alpha, \beta) \\ \int_{[\beta, \infty)} f_\pi(a, \theta)d\lambda & b = \beta. \end{cases}$$

By applying Theorem 4.6 and observing

$$\text{Var}_{\rho, \pi}(q_\pi(s, a)\widetilde{\psi}(s, T(a), \theta)) = \text{Var}_{\rho, T_*\pi}(q_\pi(s, b)\widetilde{\psi}(s, b, \theta)),$$

the proof is complete. $\qquad \square$

*Proof of Corollary 4.8.* First, it is clear $T$ is measurable. Second, $f_{MV}$ differentiable in $\theta$ implies $\Pi$ is $\lambda$-compatible. This also implies $f_{AG}$, the density of $T_*\Pi$, is differentiable in $\theta$ and therefore $T_\pi$ is $\sigma$-compatible, where $\sigma$ is the spherical measure on $(\mathcal{B}, \mathcal{F}) = (\mathcal{S}^{d-1}, B(\mathcal{S}^{d-1}))$. It is straightforward to confirm that the remainder of Conditions 4.1-4.3 hold. Applying Theorem 4.6 completes the proof. $\qquad \square$

### B.5 Policy Gradients for Parametrized Action Spaces

First we derive a stochastic policy gradient for parametrized action spaces, which we can do by writing down the policy distribution and applying 2.2. Recall a parametrized action space with $K$ discrete actions is defined as

$$\mathcal{A} := \bigcup_k \{(k, \omega) : \omega \in \Omega_k\},$$

where $k \in \{1, \ldots, K\}$.

#### Construction of a Policy Family

Masson et al. (2016) gives a definition for a policy over parametrized action spaces, and our definition is the same in spirit, but for our purposes we need to be careful in formalizing the construction. Our construction here is also a bit more general.

Informally, we can think of a policy over a parametrized action space as a mixture model, where $k \in [K]$ is a latent state. To formally define a policy family on $\mathcal{A}$, the idea will be to construct a density function $f_\pi$ that is differentiable in its parameter $\theta$. We proceed as follows:

1. Let $(\Omega_k, \mathcal{E}_k, \mu_k)$ be measure spaces.
2. For $k \in [K]$: specify $\Pi_k = \{\pi(\cdot, \theta|s) : \theta \in \Theta\}$, a $\mu_k$-compatible family of probability measures on $(\Omega_k, \mathcal{E}_k)$. Denote the corresponding densities by $f_k$.
3. Denote by $\mu_0$ the counting measure on $(\mathcal{A}_0, B(\mathcal{A}_0)) = (\mathbb{R}, B(\mathbb{R}))$, and specify $\Pi_0 = \{\pi(\cdot, \theta|s) : \theta \in \Theta\}$ a $\mu_0$-compatible family of probability measures, parametrized by $\theta_0$ and supported on $[K]$. Denote the corresponding density by $f_0$.
4. Let $\theta := (\theta_i)_i$, and define

$$f_\pi((k, \omega), \theta|s) := \begin{cases} f_0(k, \theta_0|s)f_k(\omega, \theta_k|s) & \text{if } (k, \omega) \in \mathcal{A} \\ 0 & \text{otherwise.} \end{cases}$$

To finish the policy construction, we need an appropriate $\sigma$-algebra $\mathcal{E}$ and reference measure $\mu$ such that $f_\pi$ is a measurable and $\int_\mathcal{A} f_\pi d\mu = 1$. In fact it is not difficult to construct $\mathcal{E}$ and $\mu$ in terms of $(\mathcal{E}_i)_i$ and $(\mu_i)_i$, respectively, but we do not go into detail here. Assuming such a construction exists, we can define $\Pi$ a $\mu$-compatible family of policies, parametrized by $\theta = (\theta_i)_i$.

### STOCHASTIC POLICY GRADIENT

Let $(\mathcal{A}, \mathcal{E}, \mu)$ and $\Pi$ be as constructed above. By applying Theorem 2.2, $\nabla\eta(\theta)$ can be estimated from samples by

$$g(s, a, \theta) := q_\pi(s, a)\nabla\log f_\pi(a, \theta|s) = q_\pi(s, a)\nabla\log f_0(k, \theta_0|s) + q_\pi(s, a)\nabla\log f_k(\omega, \theta_k|s).$$
$$\text{(B.6)}$$

### RESTRICTED ACTION PARAMETERS

The second term in (B.6) is simply the policy gradient for a $\mu_k$-compatible family on $(\Omega_k, \mathcal{E}_k, \mu_k)$. Let $(\mathcal{B}_k, \mathcal{F}_k)$ be a measurable space and consider the setting in which we apply a measurable function $T_k : \mathcal{A}_k \to \mathcal{B}_k$ to the action parameters before execution in the environment. Assume the conditions are satisfied to apply Theorem 4.6, and denote by $f_{k,*}$ the density of $T_*\pi_k$ with respect to an appropriate reference measure. Then we can replace $q_\pi(s, a)\nabla\log f_k(\omega, \theta_k|s)$ with $q_\pi(s, a)\nabla\log f_{k,*}(T_k(\omega), \theta_k|s)$ in (B.6) to obtain the lower variance estimator

$$\widetilde{g}(s, a, \theta) := q_\pi(s, a)\nabla\log f_\pi(a, \theta|s) = q_\pi(s, a)\nabla\log f_0(k, \theta_0|s) + q_\pi(s, a)\nabla\log f_{k,*}(T_k(\omega), \theta_k|s).$$
$$\text{(B.7)}$$

## C    DETAILS FOR APPLICATIONS

This section contains additional details on the experiments and results in Sections 5.1 and 5.2.

### C.1    2D NAVIGATION

We run each setup 24 times from a random initialization. To create the cumulative reward trajectory plots in Figure 2 we (1) use $k$-NN regression to interpolate the cumulative discounted rewards on each run, and (2) using the cumulative discounted rewards from each sample trajectory, plot the average curve with a 95% confidence band.

Table 1 gives the hyper-parameters used in the experiments on the Platform2D-v1 environment.

Table 1: Hyper-parameter settings used in training

| Hyperparameter | Setting |
|---|---|
| Num. Workers | 4 |
| Optimizer | SGD |
| Learning Rate | 0.01 |
| $\sigma$ | 0.1 |
| $\gamma$ | 0.99 |
| No. Layers: Policy Net | 2 |
| Width: Policy Net | 32 |
| No. Layers: Value Net | 2 |
| Width: Value Net | 32 |

### C.2    KING OF GLORY

Here we provide details on modeling for King of Glory, the experimental procedure and the tables referenced in Section 5.2.

#### STATE REPRESENTATION

A detailed description of all the features can be found below in Table 4. After extracting features, we take the outer product of the feature vector with itself to capture dependencies between features. To be precise, first define $\phi_0$ to be the 74-dimensional initial feature extraction. The featurized state representation $\phi(s)$ is defined by

$$(\phi(s))_{i,j} := \begin{cases} (\phi_0(s))_i(\phi_0(s))_j & \text{for } i \neq j, \\ (\phi_0(s))_i & \text{for } i = j. \end{cases}$$

By symmetry, we use only the lower triangular portion of the matrix defined above giving a $(74 \times 73)/2 + 74 = 2775$ dimensional feature vector that is input to the policy and value networks.

#### MODELING THE POLICY AND VALUE FUNCTION

The value network is modeled using a feed-forward neural network which takes as input $\phi(s)$. The sampling policy is a mixture, where the mixing distribution is over the 7 discrete actions, and a Gaussian distribution is used for each parameter space. We model the policy using 5 networks, one of which represents the distribution over the discrete actions by a fully connected feed-forward neural network into a 7-way softmax. For the parameters, we model the mean of the sampling distribution using a feed-forward network. The variance of the sampling distribution for the action parameters is $\sigma^2 \mathbf{I}$ where $\sigma$ is learned by the agent. All action parameters share the same $\sigma$ and all 5 networks share weights up to the last layer.

#### LEARNING THE POLICY

The agent is trained to play as the hero Di Ren Jie and training is against the game's internal AI, also playing as Di Ren Jie. For both methods, 10 agents are trained for 5000 episodes each. During

training the cumulative discounted reward of each episode and game outcome are tracked. The hyper-parameters we used for the neural network structure and the A3C algorithm are shown in Table 2. To construct the plots in Figure 2, we apply a low pass filter to each trajectory and then plot the average curve with a 95% confidence band.

Like Mnih et al. (2015) and others do for the Atari Learning Environment, we employ frame-skipping; two out of every three frames are skipped. Because our reward is defined in terms of a state potential function, rewards from the skipped states are still captured. For training, we use the Adam algorithm (Kingma & Ba, 2015). No parameters are shared between different networks and all networks use SElu activation functions (Klambauer et al., 2017). Table 2 contains various hyper-parameter settings we used.

Table 2: Hyper-parameter settings used in training

| Hyperparameter | Setting |
|---|---|
| Num. Workers | 8 |
| N | 128 |
| Optimizer | Adam ($\beta = (0.5, 0.9)$) |
| Actor Learning Rate | $10^{-6}$ |
| Critic Learning Rate | $10^{-3}$ |
| $\gamma$ | 0.99 |
| No. Hidden Layers: Policy Net | 2 |
| Width: Policy Net | (128,96) |
| Activation: Policy Net | SELU |
| No. Hidden Layers: Value Net | 2 |
| Width: Value Net | (128,96) |
| Activation: Value Net | SELU |

Table 3: Parametrized action space for *King of Glory*

| Action | Parameter Dimension | Description |
|---|---|---|
| no action | 0 | agent does nothing |
| move | 2 | move in direction $\omega$ |
| attack | 0 | hero uses its normal attack |
| skill 1 | 2 | hero uses skill 1 towards direction $\omega$ |
| skill 2 | 2 | hero uses skill 2 towards direction $\omega$ |
| skill 3 | 2 | hero uses skill 3 towards direction $\omega$ |
| recovery skill | 0 | hero uses the recovery skill to heal itself |

Table 4: State features for *King of Glory*

| Feature | Dimension | Range | Description |
|---|---|---|---|
| position: our hero | 2 | $[-1, 1]^2$ | x,y coordinates of our hero's position |
| position: enemy hero | 2 | $[-1, 1]^2$ | x,y coordinates of enemy hero's position |
| position: enemy hero, relative | 3 | $\mathbb{R}^3$ | distance, direction to enemy hero |
| position: enemy tower, relative | 4 | $\mathbb{R}^4$ | distance, distance relative to attack range, relative direction to the nearest enemy tower |
| position: enemy minion, relative | 3 | $\mathbb{R}^3$ | distance, relative direction to the nearest enemy minion |
| position: our spring, relative | 3 | $\mathbb{R}^3$ | distance, relative direction to our life spring |
| in tower range: our hero | 3 | $\{0, 1\}^3$ | is our hero in the range of the enemy's towers |
| in tower range: enemy hero | 3 | $\{0, 1\}^3$ | is enemy hero in the range of our tower |
| attacked by tower | 3 | $\{0, 1\}^3$ | are the enemy towers are attacking our hero |
| skill cool down: our hero | 5 | $[-1, 1]^5$ | normalized cool down time for our hero's skills |
| skill cool down: enemy hero | 5 | $[-1, 1]^5$ | normalized cool down time for enemy hero's skills |
| HP: our hero | 1 | $[-1, 1]$ | our hero's health points |
| HP: enemy hero | 1 | $[-1, 1]$ | enemy hero's health points |
| HP: nearest minion | 1 | $[-1, 1]$ | health points of the nearest enemy minion |
| HP: nearest tower | 1 | $[-1, 1]$ | health points of the nearest enemy tower |
| HP: minions in range | 1 | $\mathbb{R}^+$ | sum of HP of all the minions in the attack range of our hero |
| alive: our hero | 1 | $\{0, 1\}$ | whether our hero is alive |
| alive: enemy hero | 1 | $\{0, 1\}$ | whether enemy hero is alive |
| gold: our hero | 1 | $[0, 1]$ | our hero's gold |
| gold: enemy hero | 1 | $[0, 1]$ | enemy hero's gold |
| gold: $\Delta$ | 1 | $[-1, 1]$ | difference between our hero's gold and enemy hero's gold |
| EP: our hero | 1 | $[-1, 1]$ | normalized energy points of our hero |
| EP: enemy hero | 1 | $[-1, 1]$ | normalized energy points of enemy hero |
| hero state: our hero | 13 | $[0, 1]^{13}$ | our hero's level, experience, current money, kill count, death count, assist count, total money, attack range, physical attack, magical attack, move speed, health points, energy points |
| hero state: enemy hero | 13 | $[0, 1]^{13}$ | enemy hero's level, experience, current money, kill count, death count, assist count, total money, attack range, physical attack, magical attack, move speed, health points, energy points |

Table 5: State features and weights used in reward design

| Feature | Weight | Description | Notes |
|---|---|---|---|
| gold difference | 0.5 | difference between the amount of our hero and enemy hero | |
| HP (our hero) | 0.5 | health points of our hero | |
| hurt to enemy hero | 0.5 | total amount of hurt from our hero to enemy hero | |
| hurt to enemy | 1.0 | total amount of hurt from our hero to all the enemies | |
| kill dead difference | 1.0 | difference between kill count and dead count | |
| distance to our life spring | $0.25 \times (1.0 - \text{HP})$ | distance from our hero to spring | $\text{HP} \in [0, 1]$ |
| distance to enemy | $0.125 \times \text{HP}$ | distance from our hero nearest enemy | $\text{HP} \in [0, 1]$ |
| tower HP difference | 1.0 | difference between HP of our tower and enemy tower | |
| crystal HP difference | 2.0 | difference between HP of our crystal and enemy crystal | |
| skill hit rate | 0.15 | percent of emitted skills that hit enemy hero | |
| win/loss | 2.0 | game result | |

