# OpenReview forum: "Marginal Policy Gradients: A Unified Family of Estimators for Bounded Action Spaces with Applications"
_ICLR.cc/2019/Conference_

### Official Review · AnonReviewer1 · 2018-11-02
**Fun, albeit incremental paper**

**Rating:** 7
**Confidence:** 4

**Review:**

Summary

This paper derives a new policy gradient method for when continuous actions are transformed by a
normalization step, a process called angular policy gradients (APG). A generalization based on
a certain class of transformations is presented. The method is an instance of a
Rao-Blackwellization process and hence reduces variance.


Detailed comments

I enjoyed the concept and, while relatively niche, appreciated the work done here and do believe it has clear applications. I am not convinced that the measure theoretic perspective is always
necessary to convey the insights, although I appreciate the desire for technical correctness. Still,
appealing to measure theory does reduces readership, and I encourage the authors to keep this in
mind as they revise the text.

Generally speaking it seems like a lot of technicalities for a relatively simple result:
marginalizing a distribution onto a lower-dimensional surface.

The paper positions itself generally as dealing with arbitrary transformations T, but really is
about angular transformations (e.g. Definition 3.1). The generalization is relatively
straightforward and was not too surprising given the APG theory. The paper would gain in clarity
if its scope was narrowed.

It's hard for me to judge of the experimental results of section 5.3, given that there are no other
benchmarks or provided reference paper. As a whole, I see APG as providing a minor benefit over PG.

Def 4.4: "a notion of Fisher information" -- maybe "variant" is better than "notion", which implies there are different kinds of Fisher information
Def 3.1 mu is overloaded: parameter or measure?
4.4, law of total variation -- define


Overall

This was a fun, albeit incremental paper. The method is unlikely to set new SOTA, but I appreciated
the appeal to measure theory to formalize some of the concepts.


Questions

What does E_{pi|s} refer to in Eqn 4.1?
Can you clarify what it means for the map T to be a sufficient statistic for theta? (Theorem 4.6)
Experiment 5.1: Why would we expect APG with a 2d Gaussian to perform better than a 1d Gaussian
on the angle?


Suggestions

Paragraph 2 of section 3 seems like the key to the whole paper -- I would make it more prominent.
I would include a short 'measure theory' appendix or equivalent reference for the lay reader.

I wonder if the paper's main aim is not actually to bring measure theory to the study of policy
gradients, which would be a laudable goal in and of itself. ICLR may not in this case be the right
venue (nor are the current results substantial enough to justify this) but I do encourage authors to
consider this avenue, e.g. in a journal paper.

= Revised after rebuttal =

I thank the authors for their response. I think this work deserves to be published, in particular because it presents a reasonably straightforward result that others will benefit from. However, I do encourage further work to
1) Provide stronger empirical results (these are not too convincing).
2) Beware of overstating: the argument that the framework is broadly applicable is not that useful, given that it's a lot of work to derive closed-form marginalized estimators.

---

> ### Author Response · Authors · 2018-11-15
> **Authors' Response to Reviewer 3**
>
> Thank you for the time and effort spent reviewing our paper, and for the detailed suggestions. Below we repeat the questions/comments from the review and respond to each in turn.
>
> “The paper positions itself generally as dealing with arbitrary transformations T, but really is about angular transformations (e.g. Definition 3.1). The generalization is relatively straightforward and was not too surprising given the APG theory. The paper would gain in clarity if its scope was narrowed.”
>
> Our MPG framework not only supports the angular transformation but also covers the recently proposed clipped transformation in CAPG [Fujita and Maeda, 2018]. The theoretical result is tighter than the one in [Fujita and Maeda, 2018], and it supports general transformations instead of only clipped actions.
>
> "I am not convinced that the measure theoretic perspective is always necessary to convey the insights, although I appreciate the desire for technical correctness." / "Generally speaking it seems like a lot of technicalities for a relatively simple result: marginalizing a distribution onto a lower-dimensional surface."
>
> We agree that the measure theoretic approach is not always necessary (indeed for angular actions, it is not needed), but it is necessary for a very common scenario -- clipped actions. Researchers and practitioners both almost always clip actions when using policy gradient algorithms for robotics control environments (read: MuJoCo tasks). Recently, a reduced variance method was introduced by Fujita and Maeda (2018) for clipped action spaces. Their algorithm is also a member of the marginal policy gradients family and our theoretical results for MPG significantly tighten the existing analysis of that algorithm.
>
>
> "It's hard for me to judge of the experimental results of section 5.3, given that there are no other benchmarks or provided reference paper. As a whole, I see APG as providing a minor benefit over PG."
>
> For the results in Section 5.3, the issue is that currently, there are no benchmark environments for directional control. We anticipate that in the future this may change (e.g. console and PC games often have directional controls).
>
> “What does E_{pi|s} refer to in Eqn 4.1?”
>
> The expectation is taken with respect to the policy \pi conditioned on the current state s (s here is arbitrary, but fixed). Stated differently, we are taking the expectation with respect to the distribution $\pi(\cdot | s,\theta)$.
>
> “Can you clarify what it means for the map T to be a sufficient statistic for theta? (Theorem 4.6)”
>
> We have now removed this part of the statement because we are no longer absolutely certain of its correctness, and because it is not used anywhere else in the paper.
>
> “Experiment 5.1: Why would we expect APG with a 2d Gaussian to perform better than a 1d Gaussian on the angle?”
>
> Because using a 1D Gaussian requires either (1) clipping the angle to [0,2\pi) before execution in the environment and making updates using the clipped output or (2) using the sampled angle for updates and perform the clipping in the environment. In the first case, this approach is asymmetric in that does not place similar probability on $\mu_{\theta}(s) - \epsilon$ and $\mu_{\theta}(s) + \epsilon$ for $\mu_{\theta}(s)$ near to $0$ and $2\pi$. In the second case, this requires approximating a periodic function. We include both these reasons at the start of Section 3.
>
>
> Lastly, thank you for the concrete suggestions:
> "Def 4.4: "a notion of Fisher information" -- maybe "variant" is better than "notion", which implies there are different kinds of Fisher information
> Def 3.1 mu is overloaded: parameter or measure?
> 4.4, law of total variation -- define "
>
> We have addressed these and uploaded a new draft to reflect the changes. For the last suggestion, we currently define the law of total variance(variation) in the preliminaries so we did not repeat the definition in Section 4.4. We now write "law of total variance" instead of "law of total variation" to avoid any ambiguity.

---

### Official Review · AnonReviewer2 · 2018-11-05
**Limited setting of directional RL, but interesting approach and results.**

**Rating:** 6
**Confidence:** 3

**Review:**

This paper introduces policy gradient methods for RL where the policy must choose a direction (a.k.a., the navigation problem).

Mapping techniques from "non-directional" problems (where the action space is not a direction) and then projeting on the sphere is sub-optimal (the variance is too big). The authors propose to sample directly on the sphere, using the fact that the likelyhood of an angular Gaussian r.v. has *almost* a closed form and its gradient can almost be computed, up to some normalization term (the integral which is constant in the standard Gaussian case).


This can be seen as a variance reduction techniques.

The proofs are not too intricate, for someone used to variance reduction (yet computations must be made quite carefully).


The result is coherent, interesting from a theoretical point of view and the experiment are somehow convincing. The main drawback would be the rather incrementality of that paper (basically sample before projecting is a bit better than projecting after sampling) and that this directional setting is quite limited...

---

> ### Author Response · Authors · 2018-11-15
> **Authors' Response to Reviewer 2**
>
> Thank you for the time and effort spent reviewing our paper. We mostly agree with your characterization of our work, but we think there are two important points we perhaps did not sufficiently emphasize in our paper and that we would like to mention:
>
> (1) There are other existing tasks and algorithms that fall into the marginal policy gradients framework. For example, researchers and practitioners both almost always clip actions when using policy gradient algorithms for robotics control environments (read: MuJoCo tasks). Recently, a reduced variance method was introduced by Fujita and Maeda (2018) for clipped action spaces. Their algorithm is also a member of the marginal policy gradients family and our theoretical results for MPG significantly tighten the existing analysis of their algorithm.
>
> (2) To the best of our knowledge, our work is the first to apply such variance reduction techniques to RL.
>
> To summarize, our work consists of two components: (a) a new algorithm for directional control and (b) a variance reduction framework that can be applied to directional action space and clipped action spaces. While directional action spaces are not very common at this time, clipped action spaces are extremely common. We also anticipate that in the future, many additional environments will be available that feature directional actions (many console or PC games, for example). For these reasons, we feel that our work is not incremental at all, and is actually quite novel.

---

### Official Review · AnonReviewer3 · 2018-11-05
**Comprehensive analysis and evaluated algorithms on realistic experiments.**

**Rating:** 7
**Confidence:** 3

**Review:**

In this paper the authors proposed a new policy gradient method, which is known as the angular policy gradient (APG), that aims to provide provably lower variance in the gradient estimate. Here they presented a stochastic policy gradient method for directional control. Under the set of parameterized Gaussian policies, they presented a unified analysis of the variance of APG and showed how it theoretically outperform (in terms of having lower variance) than other state-of-the art methods. They further evaluated the APG algorithms on a grid-world navigation domain as well as the King of Glory task, and showed that the APG estimator significantly out-performs the standard policy gradient.

In general I think this paper addressed an important issue in policy gradient in terms of deriving a lower variance gradient estimate. In particular the authors showed that under the parameterized marginal distribution, such as the angular Gaussian distribution, the corresponding APG estimate has a lower variance estimate than that of CAPG. Furthermore, I also appreciate that they evaluated these results in realistic experiments such as the RTS game domains.

My only question is on the possibility of deriving realistic APG algorithms beyond the class of angular Gaussian policy. In terms of the layout of the paper, I would also recommend including the exact algorithm pseudo-code used in the main paper.

---

> ### Author Response · Authors · 2018-11-15
> **Authors' Response for Reviewer 3**
>
> Thank you for the time and effort spent reviewing our paper. We are glad you liked the paper. We want to emphasize one point that we perhaps did not highlight enough in our paper: there are other existing algorithms that fall into the marginal policy gradients framework. Specifically, researchers and practitioners both almost always clip actions for use in robotics control environments (read: MuJoCo tasks). Recently, a reduced variance method was introduced by Fujita and Maeda (2018) for clipped action spaces. Their algorithm is also a member of the marginal policy gradients family and our theoretical results for MPG significantly tighten existing analyses of variance reduction that can be achieved for clipped actions.
>
> To respond to your question, yes it is possible (e.g. the example given above), but their is no general procedure that we know of to derive such methods. Rather, this would be done on an action space by action space basis

---

> > ### Comment · AnonReviewer3 · 2018-12-10
> > **Thank you for your response**
> >
> > After reading the rebuttal and other reviews, I would keep my original scores and think this paper presents some simple (of clipping action spaces and marginalizing distributions to lower dimensions and to take gradients) but very useful results in reducing variance of RL methods with continuous action spaces.  To me, this idea is worth publishing at ICLR.
> >
> > However, with limited effort/time spent on reviewing the theoretical results in Appendix, I unfortunately cannot justify the correctness of the theoretical results, nor argue whether this machinery is a must for proving this simple method.

---

### Meta-Review · Area_Chair1 · 2018-12-11
**Variance reduction for Policy Gradients with meaningful theory**

**Confidence:** 4
**Recommendation:** Accept (Poster)

**Metareview:**

The paper introduces a new variance reduced policy gradient method, for directional and clipped action spaces, with provable guarantees that the gradient is lower variance. The paper is clearly written and the theory an important contribution. The experiments provide some preliminary insights that the algorithm could be beneficial in practice.